# Digital Transformation in Higher Education Institutions: A Systematic Literature Review

**DOI:** 10.3390/s20113291

**Published:** 2020-06-09

**Authors:** Lina María Castro Benavides, Johnny Alexander Tamayo Arias, Martín Darío Arango Serna, John William Branch Bedoya, Daniel Burgos

**Affiliations:** 1Department of Industrial Engineering, Faculty of Engineering and Architecture, National University of Colombia, Manizales 170003, Colombia; jatamayoar@unal.edu.co; 2Department of Information and Documentation Science, Library Science and Archival Science, Faculty of Human Sciences, University of Quindío, Armenia 630004, Colombia; 3Department of Engineering of the Organization, Faculty of Mines, National University of Colombia, Medellín 050034, Colombia; mdarango@unal.edu.co; 4Department of Computer Science and Decision, Faculty of Mines, National University of Colombia, Medellín 050034, Colombia; jwbranch@unal.edu.co; 5Research Institute for Innovation & Technology in Education (UNIR iTED), Universidad Internacional de La Rioja (UNIR), 26006 Logroño, Spain; daniel.burgos@unir.net

**Keywords:** systematic literature review, digital transformation, digitalization, university, higher education institution

## Abstract

Higher education institutions (HEIs) have been permeated by the technological advancement that the Industrial Revolution 4.0 brings with it, and forces institutions to deal with a digital transformation in all dimensions. Applying the approaches of digital transformation to the HEI domain is an emerging field that has aroused interest during the recent past, as they allow us to describe the complex relationships between actors in a technologically supported education domain. The objective of this paper is to summarize the distinctive characteristics of the digital transformation (DT) implementation process that have taken place in HEIs. The Kitchenham protocol was conducted by authors to answer the research questions and selection criteria to retrieve the eligible papers. Nineteen papers (1980–2019) were identified in the literature as relevant and consequently analyzed in detail. The main findings show that it is indeed an emerging field, none of the found DT in HEI proposals have been developed in a holistic dimension. This situation calls for further research efforts on how HEIs can understand DT and face the current requirements that the fourth industrial revolution forced.

## 1. Introduction

### 1.1. Rational

Digital transformation (DT) has become a priority for higher education institutions (HEIs) in this second decade of the 21st century, and this is a natural and necessary process for organizations that claim to be leaders of change and be highly competitive in their domain. Several authors have defined the digital transformation from the field of business. Among them [1], who express that digital transformation is concerned with the changes digital technologies can bring about in a company’s business model, which result in changed products or organizational structures or in the automation of processes. Later, [2] defines, “Digital transformation is the profound transformation of business activities and organizations, processes, competencies and models, for the maximum transformation of the changes and opportunities of a technology mix and its accelerated impact on society, in a strategic and prioritized way.”

If HEIs want to persist in time as a key element of this transformation, and not disappear from the stage, it is necessary that they evolve integrally. Moreover, exploiting efficiently all the opportunities and potentialities opened up by the wealth of digital technologies available, redefining complete business models across the entire value chain is not straightforward and, for sure, is a challenging task. This challenge is more pressing for organizations that permanently try to assure they have competitive positioning in a global market, but the same concern is becoming pertinent for universities, as competition to select the best students and researchers is increasing [3]. Remarkably, HEIs face a disruptive scenario that is established in the new business models, ostensibly transforming the way they evolved over time, actively linking internal and external clients, and increasing their commitment and strengthening their experience in the organization [4]. Nevertheless, many universities are developing specific digital strategies in reaction to the massive shift towards using new technology, yet lack the vision, capability, or commitment to implement them effectively [5]. In this sense, it is important to have a comprehensive vision of the whole DT in HEIs, in order to obtain an overview of the current state of the art in DT in HEIs, and determine its distinctive characteristics as dimensions, actors, and implementations, which have taken place in the process of digital transformation within HEIs. 

This paper provides an overview of the research work reported in the field by means of a systematic literature review (SLR) of DT in HEIs. The sections of the paper are structured as follows: The introduction section makes out the current SLR related to the DT in HEIs, defining the objectives and the research questions. In the Methodology section, the authors state the protocol followed, the process for extracting the relevant data, and describe results of the data extracted process. The Discussion section offers a debate in order to answer the research questions. The Risk and Validity section presents the risks intrinsic in the SLR. Finally, in the Conclusions section, inferences are described.

### 1.2. Review Questions

It was used PICO criteria (population, intervention, comparison, outcomes) to identify keywords and defined search strings from research questions.

*Population*: HEIs.

*Intervention*: DT processes at HEIs.

*Comparison*: In this study no comparison intervention has been projected.

*Outcomes*: Distinctive characteristics of the DT implementation process that have taken place in the HEIs.

*Timing*: Between 1980 and April 2019 was selected for inclusion, because the year 1983 is considered as a starting point recognizing the birth of the Internet as one of the foundations of the DT.

*Setting*: Restrictions by document type, articles, and conference proceedings were analyzed.

*Language*: Articles were reported in the English in order to avoid bias, by recognizing this language as the universal language.

The aim of this SLR is to summarize the existing evidence on main distinctive characteristics of the DT implementation process that has taken place in the HEIs.

According to [6], the following explicit statement of the research question (RQ), which the review will address with reference to participants, interventions, and outcomes is: RQ—What are the main distinctive characteristics of DT implementation process that have taken place in the HEIs?

### 1.3. Identification of the Need for a Review

To ascertain if there were previous studies such as the one proposed in this article, authors considered the following search: (“digital transformation*”) AND (“higher education institution*” OR universit*) AND (“systematic literature review” OR “SLR” OR “systematic mapping”).

As a result, for this search, no SLR articles were found. As it is important to have a comprehensive vision of the whole DT in HEIs, in order to obtain an overview of the current state of the art in DT in HEIs, and determine the dimensions, actors, implementations that have taken place in the process of digital transformation within HEIs, the authors considered it feasible to undertake a systematic literature review about DT in HEIs.

## 2. Methodology

This research was supported following the [7] protocol to demand that this is a comprehensive, objective, and reliable overview, and not a partial review of a convenience sample. The steps in the systematic literature review method are documented below.

### 2.1. Data Sources

The search was carried out through the electronic databases Web of Science (WoS), and Scopus, as they are the most relevant scientific information platforms that access the scientific databases and the most significant publications of the different areas of knowledge. In particular, regarding issues of digital transformation in higher education institutions.

Both databases allow advanced structures to be searched using logical operators that conform to the specifications of the systematic review proposed in this research. As well as the tools of filtration and bibliometric analysis that provides valuable information to the systematic review proposed in this research.

### 2.2. Search Strategy

One of the most subtle, but relevant moments of an SLR is the structured search strategy, because it must allow filtering the information available in the databases, so that the selected articles will respond to the questions raised in the investigation, and consequently the stated objective will be fulfilled. The search strategy must allow the completeness of the search to be assessed [8].

In response to this requisite, the words contained in the search strategy, the keywords considered in the PICO model, and also the research questions, were identified

The structured search used to realize the search of the articles was (“digital transformation*”) AND (“higher education institution*” OR Universit*) and was conducted on 10 April 2019. 

The search string adapted to the syntax required by the Institute for Institute for Scientific Information–Scopus database was as follows: (TITLE-ABS-KEY (“digital transformation*”) AND TITLE-ABS-KEY ((“Higher Education Institution*” OR universit*))) AND DOCTYPE (ar OR cp) AND PUBYEAR > 1979 AND PUBYEAR < 2020 AND (LIMIT-TO (LANGUAGE, “English”)).

The search string adapted to the syntax required by the Institute for Scientific Information–Web of Science database was as follows: SUBJECT: ((“Digital transformation *”)) AND SUBJECT: ((“Higher Education Institution*” OR Universit *)). Refined by: LANGUAGES: (ENGLISH) AND TYPES OF DOCUMENTS: (ARTICLE). Period of time: Every year. Indices: SCI-EXPANDED, SSCI, A & HCI, ESCI.

### 2.3. Study Selection

Studies were selected according to the criteria outlined below.

#### 2.3.1. Study Selection Criteria

Authors [7] explained that a study identified by electronic and hand searches, can be clearly excluded based on title and abstract, and a full copy should be obtained. Additionally, duplicate references were eliminated using Mendeley software. 

#### 2.3.2. Study Selection Process

Studies were selected according to the criteria outlined below:

*Study designs*: We included studies where it could be identified which dimensions of HEIs have been permeated by digital transformation, who has intervened in these processes, their methodology, technologies adopted, among others.

Taking into account the suggestion given by [7] using PICO criteria, to identify keywords and defined search strings from the research question: 

Population: HEIs.

*Intervention*: DT processes at HEIs.

*Comparison*: In this study no comparison intervention was projected.

*Outcomes*: Distinctive characteristics of DT implementation processes that took place in the HEIs.

*Timing*: Between 1980 and April 2019 was selected for inclusion, because the year 1983 is considered as a starting point recognizing the birth of the Internet as one of the foundations of DT.

*Setting*: Restrictions by document type, articles and conference proceedings were analyzed.

*Language*: Articles reported in English in order to avoid bias, by recognizing this language as the universal language.

### 2.4. Study Quality Assessment

The paper’s compliance with the inclusion/exclusion criteria were verified by the reviewers following the conditions outlined below.

#### 2.4.1. Study Selection Criteria

##### Eligibility Criteria

First, article included in its title the “digital transformation” sequence words AND, and second, articles included in its abstract the “digital transformation” sequence words AND, either HEIs or university.

It was marked with number 1, if the word appeared, and marked with number 0, if it did not. In cases where the title and abstract were not enough to decide, authors assessed the entire content of the paper. In order to classify if a paper fulfilled these criteria the following logic operation took place. In cases where the title and abstract were not enough to make a decision; the authors assessed the entire content of the paper. In order to classify if a paper fulfilled these criteria, the following logic operation took place.

IF(AND(TITLE=1;ABSTRACT=1; COUNT.IF(ABSTRACT:ABSTRACT;1)>=1);”candidate articles”;“no”).

##### Inclusion/Exclusion Criteria

One of the tasks described as critical in [7] was the quality assessment criteria, and therefore was considered as a challenge in a systematic review. For this purpose, they constructed a quality questionnaire based on 5 issues affecting the quality of the study, which were scored to provide an overall measure of study quality. 

In that sense, questions were adapted to the current study, and the following questions are relevant insofar as they define the inclusion/exclusion of articles for full reading and subsequent analysis. That is, rather than just considering dedicated solutions, platforms or system of systems, the paper should consider the evolutionary characteristics of the digital transformation. 

To minimize study bias and maximizes internal and external validity authors constructed a quality questionnaire based on [7] 

The questions have been classified in the following categories. 

Study design. Articles that demonstrate the objective and the process of the DT that was carried out inside the HEI.Are all research questions answered adequately?Are the main goals of the DT at HEIs stated?Does the paper outline the methods used to address DT in HEIs?System design. Articles that show the dimensions, participants, and/or their relationships in processes of DT of HEIs.
Does the proposed DT apply to the whole HEI?Are the business model, dimensions, technology, actors, and relationship involved in DT at HEIs clearly described and defined?Were all model construction methods to apply DT in HEIs fully defined?

The quality assessment checklist described the score according to the level of article´s quality. Table 1 quality assessment score assigned to each question according to the information details provided regarding the topic of DT in HEIs.

Papers were included and classified as “Full reading article” in the next stages if the sum of the criteria were greater than 4 points.

### 2.5. Data Extraction

#### 2.5.1. Design of Data Extraction Forms

The software used to manage the data and analyze articles information, and the reference manager, were Mendeley and Microsoft Excel. 

Mendeley was used to manage the articles resulting from the search in the scientific databases, to eliminate duplicate references and to classify the information from each article, underlining it with a different color according to the category.

On the other hand, to document and manage data resulting from the following protocol we used Excel. The workbook was made up by several tabs, where each phase was documented in them.

The information is available in the Appendix A.

#### 2.5.2. Data Extraction Procedures

This data collection process was developed in three stages.

*Information Analysis*: The analysis and classification of the article’s information was a bottom-up analysis. The text fragments that answer the research questions were highlighted with different colors, using the Mendeley tool. This action allowed for further reading and detailed analysis and classification.

*Classification of Information*: Label codes to assign representative meaning to the highlighted information were defined synchronously with the Information Analysis stage. Table 2 shows the codes considered for each of the research questions.

*Information Extraction*: Each of the text fragments highlighted in Information Analysis stage were classified according to the codes established in the Classification stage. A spreadsheet was required to managing the resulting information of this stage.

https://drive.google.com/file/d/17Ovoq4OibkzWFJMzxizonWGu2SmsTWcl/view?usp=sharing.

### 2.6. Data Syntesis

The data was tabulated and displayed to represent:

Different DT in HEIs definitions were presented in the articles. 

Dimensions within a HEI that have established the DT or have been forced to intervene in DT processes.

Actors, methods, goals, and technologies that became more relevant in the DT in HEIs from the social, organizational, and technological perspectives.

Actors that were involved in the DT of HEIs processes.

Routes established by HEIs to carry out their DT.

## 3. Results

This section is structured in response to research and mapping questions consolidated from evaluating the articles once the information analysis process described in the review protocol had taken effect. Further, data extracted from the review protocol are consolidated in the spreadsheet:

https://drive.google.com/file/d/17Ovoq4OibkzWFJMzxizonWGu2SmsTWcl/view?usp=sharing. 

### 3.1. Included and Excluded Studies

This subsection presents the evolution of the number of records in the SLR phases of the protocol.

#### Search Strategy

Once the search strategy was executed in the scientific databases, the following Table 3 presents the records obtained.

It is necessary to clarify that 119 records were identified through database searching, and the search on the databases was conducted on 10 April 2019.

*Study selection criteria:* As a result, after duplicates and incorrect titles and abstracts were removed there were selected 106 records and 66 excluded were in this section. 

*Study selection process:* The number of elected papers once the eligibility criteria process took place are described in Table 4.

Inclusion/Exclusion criteria. After applying inclusion and exclusion criteria, the number of papers selected to full reading are described in Table 5.

Figure 1 presents the protocol phases and the evolution of the number of records in each one of them.

### 3.2. Definitions of DT that Are Stated in the Literature and Are Applied to HEIs

Definitions of DT that are stated in the literature and are applied to HEIs are shown in Table 6.

### 3.3. How Has the DT of HEIs Been Addressed?

The number of papers on digital transformation in HEIs evolved over time has increased significantly. It can be appreciated in Figure 2 that since 2016, the number of publications has increased annually by 200%, and by March 2019, the increase was already 133%.

As seen in Figure 3, from 2016 to date, research articles have addressed the digital transformation in HEIs from technological, organizational, and social perspectives. This is how the great interest from the technological perspective was seen during 2017, where the percentage of papers that addressed the digital transformation in HEIs from that perspective was 67%. In 2018, research interest increased from the social and organizational perspectives, increasing from 17% in 2017 to 39% in 2018. By now, in the year 2019, the trend for the social perspective denotes great importance and interest covering 57% of the papers analyzed, and 29% of the researches were dedicated to the organizational perspective, leaving in last place the technological perspective with 14%.

Below, is Figure 4, showing the radial scheme of the dimensions, which within a HEI, have received the DT or have been forced to intervene in DT processes. Teaching has been the dimension most influenced by technologies intervention, while the least addressed has been the marketing dimension.

### 3.4. Interrelationships inside DT of HEIs

To generate the existing relationships between dimensions of DT in HEIs, Gephi software was used. Authors [21] define Gephi as an open source network exploration and manipulation software, which uses a 3D render engine to display large networks in real-time and to speed up the exploration. A flexible and multi-task architecture brings new possibilities to work with complex data sets and produce valuable visual results. 

In general, the concepts that have been defined in Table 6 are represented by the nodes, and the relationships that exist between them are represented by their closeness to each other, and by the edges. Additionally, the thickness and intensity of the color of the edge indicates the frequency of appearance of the concept in the analyzed articles. In the spreadsheet contained in the following link, the normalized data that was executed by the Gephi software can be seen. https://drive.google.com/file/d/1nFO_bRKmkLTF2yryqFoG3wHT-jlCBRWn/view?usp=sharing


In Figure 5, it can be seen that the actors became more relevant in the DT in HEIs from the social, organizational, and technological perspectives, and important actors revolve according to their influence on the relationship. 

Figure 6 illustrates which DT guidelines are required to carry out DT in HEIs from the social, organizational, or technological perspective. A strategy that guides the DT in all perspectives is required. The strategy should be designed from a holistic perspective of the DT in the HEI.

Figure 7 shows the goals leading the HEI to undertake DT processes inside it. This view allows us to ascertain the great variety of objectives that have led the HEI to immerse themselves in the DT processes. It is important to note that depending on the perspective being addressed, your goals can change.

Figure 8 shows that the diversity of technologies used in the DT process in HEIs depends on the social, organizational, or technological perspective with which it has been approached. Likewise, as in the previous figure, the technologies used depended largely on the perspective addressed.

### 3.5. DT of HEIs Addressed by Actors

It was observed that 95% of the articles considered that the actors present in a process of DT were students and teachers. Fifty-three percent of them link industry as an actor that must be considered when initiating a DT in HEIs. Forty-seven percent of the articles include as important actors of the university managers process, in addition to having a DT Team. Further, 42% of them establish, as an important element, the government, both internal, local and national, and the organic units within the HEI. Likewise, 32% of the articles relate to graduates and researchers. Moreover, 26% of the articles also raise community as an actor to be taken into account, as well as the faculty members, and digital platforms. Furthermore, 21% of the articles consider that there must be an IT business leader. Additionally, 16% of the articles include as an important element in the DT process in HEIs, namely the teacher training unit. Similarly, 11% of the articles linked the departments, schools, and parents as important actors, as well as the existence of content providers or information system. Finally, 5% of the articles express the importance of the rectory in DT processes of the HEI. Figure 9 shows actors that took place in a DT in a HEI process.

### 3.6. Route Established by HEIs to Carry Out their DT

Figure 10 presents the diverse implementing methods of DT at a HEI. Overall, 68% of articles considered that to begin the process of DT of an HEI, a guideline is required to route its implementation. Forty-seven percent of the articles agree that DT must pay close attention to the digitalization process. This is supported with 37% that include the creation of DT centers. Forty-two percent of the articles refer to the creation of a competence center to align the human resource with DT in the HEI. On the other hand, although 37% raise the need to link a re-engineering process in HEIs, only 16% took into account the implementation of an enterprise architecture, and 11% of the IT architecture management. Additionally, 26% understood the necessity of the build and running system according to the requirements of the HEI. Finally, 26% expressed the importance of considering the change management as a vital strategy to success in a HEI DT.

## 4. Discussion

In this section we discuss the findings related to our main research question based on the results obtained.

### 4.1. Main Distinctive Characteristics of DT in HEIs

#### DT at HEIs Reflected in the Literature

It is remarkable to note that DT within HEIs has been approached from different perspectives and a consensus on its definition has not yet been consolidated. Researchers [22] introduced DT as an element disruptor that fundamentally changes entire industries and organizations. While, researchers [23] recognized that digitized organizations need to focus on both technology domain and social domain for a successful transformation. Moreover, researchers [10,14,19,20,24] consider that DT in HEIs from a renewal business model perspective are aligned with technological trends. Additionally, researcher [10] added elements involved in the DT process such as people, processes, strategies, structures, and competitive dynamics. On the other hand, researchers [12,17] involved a social aspect that intervenes in the DT process, aiming at the transformation towards the customer experience lifecycle and how DT improves or replaces traditional products and services. In addition, researchers [16] linked DT as a resource to create additional and differentiated value and extended the spectrum of DT in HEIs to interactions across company borders with clients, competitors, and suppliers. Finally, [18,25] observed DT from the educational dimension, integrating digital technologies in teaching, learning, and organizational practices. 

### 4.2. How Has the DT in HEIs Been Addressed?

The emphasis on the implementation of DT processes by the HEIs depends on their interest and necessities, which is why the tendency has evolved over time, from the technological perspective, then organizational, to finally consolidate in the social perspective. First, the education dimension is the one that has been most permeated by the DT processes in the HEI, followed by the infrastructure, and then the curriculum and business administration. This is how each of these perspectives have addressed various dimensions that in themselves, have a variety of application fronts. Below is detailed each dimension presented in Figure 4.

#### 4.2.1. DT in HEIs Dimensions

Teaching dimension: The DT seen from the teaching dimension has several fronts. 

Digital platforms and contents for teaching and learning: Authors [26] considered that it was important to use the tools which satisfy contemporary educational standards and methods, first of all, the tools based on digital technology [24]. 

Innovate pedagogical methodologies: Authors [27] emphasized that innovations in digital teaching are not just technical innovations, but rather academic, curricular, organizational and structural innovations. In this respect, the use of digital educational resources is perceived as enabling new roles for teachers and students, creating flexible and motivating ways of learning, being more autonomous and collaborative [18]. 

Digital literacy and digital skills: In the digital economy, the necessity arises for new highly professional work force with digital skills and competence in the technology and communication field [28]. Moreover, researchers [27] expressed that in terms of university teachers’ perspectives, technical as well as pedagogical guidance, is recommended. 

Teaching administration process: From an administrative level, authors [29] reflected that many HEIs have leveraged the use of technology to provide flexibility in learning and just-in-time training for learners in the efforts to improve both the internal processes of course delivery and enhance the provisions of education quality. 

Infrastructure dimension: The DT seen from the infrastructure dimension has diverse frontages, depending on the dimension that support. 

Digital infrastructure for teaching: Digital platforms and learning platforms are important tools which satisfy contemporary educational standards and methods, according to the authors [23,26,30]. Physical infrastructure for teaching, as laboratories. For instance, [31] learning factory, and [23] living lab.

Data and security infrastructure: Author [10] expressed that with the increased use of digital technologies and the growing connectivity of everything come also greater challenges on the level of security, compliance and data protection, and regulations.

Software infrastructure for HEIs: Researchers [16,18,25,29,32] contemplated an agile platform and flexible architecture that could handle adaptive and emergent processes (administrative, teaching, and researching).

Curriculum dimension: This dimension has several views depending on the DT process that has been performed.

Curriculum modernization: Modernize the curricula which satisfy contemporary educational standards and methods, developing international curricula, finding new ways of delivering content through digital learning and the widening use of ICT technologies [18,26,27]. Flexible curriculum: A flexible response to the needs of labor market is the main goal of updating the educational program [28].

Digital curriculum: Author [10] concluded that students are increasingly demanding an improvement in the “basics” of their experience, with features such as digitization of administrative processes, unrestricted 24-hour access to all information, and services using multiple platforms or digital curriculum. 

Administration dimension: Improve existing work and operations: Actually, HEIs are using DT strategies to improve “how” they do their existing work, to apply changes in value creation, to reflect the influence of DT, while building new digital models in parallel, or fully digitizing their current considering the new demands of the labor market and the growing expectations of students to innovate their experiences regarding learning, teaching, research and management [10,16].

Financial and technological aspects: The DT requires large investments to get rid of the past and adopt new and extreme technologies [10,16,19].

Reorganize administrative units: To succeed, universities must re-structure their model of academic and administrative governance to act quickly and precisely, efficiently developing new concepts, and enabling a flexible and supportive infrastructure, starting with a mindset change towards an “entrepreneurial mindset” [19,32].

Making Informed Decisions: Activity reports and service level indicators integrated with business intelligence mechanisms, provide a comprehensive vision of the on-going business processes, and a critical view for effective decision making [16].

Research dimension: Research is forced to align with the DT to fulfill with requirement and expectations of the actors involved in the research processes [10]. 

Human resource dimension: There is a bidirectional relationship between DT in HEIs and human resources. On the one hand, DT influences and impact human resource factors and contributes to enhancing productivity [14]. On the other hand, digital capabilities of human resources are the key enabler of university DT through the competent digital workforce [20].

Extension dimension: The use of independent certification of competence and the establishment of integration links between universities, specialized secondary schools, major enterprises, and public administration in the region [12,28,33].

DT governance dimension: For highly digitized organizations, understanding and managing digital innovations is crucial, as any change can be an important factor in successful implementation [12,23]. Likewise, the educational organization needs analysis and should be aligned with, and within, the scope of the governance strategy and management model of higher education, taking into account the different normative and non-normative scenarios, as well as, the theory of corporate governance, which must correspond to theoretical, organizational, and strategic aspects of innovative resource allocation [16,18,19,28]. Furthermore, this implies taking into account the risk management to minimize the impact of these innovations on members of the organization [22,23].

Information dimension: Information dimension in the DT in HEIs is a very valuable asset, therefore it must be aligned and be consistent with the business architecture of the HEI. First, in order to enrich the internal process of strategic formulation and implementation, data from various sources can be streamlined for a leaner and more effective planned business [22,25]. Second, from teaching perspective, in modern conditions, educational materials are already created in digital formats, and become the key enabler of online education [20].

Marketing dimension: Marketing dimension is integrated in the DT in HEIs as a new facet of the HEI that requires a digital marketing model [19,30]. 

Business process dimension: DT promotes the re-invention of the institution the transition from related operational procedures to the use of digital technologies to improve, enhance, or replace traditional services with digital ones, to simplify the processes involved in educational service delivery and operational complexity [12,16,17,19,22,25,27,29,30].

#### 4.2.2. Remarkable Relationships of DT of HEIs

The most notable relationships established in DT in HEIs that have been established after analyzing the information are specifically related to the actors involved, the goals that guide DT processes, the employed methods, and the technologies that have been used. Below each one is described.

Actors: Although the most important actors involved depend on each perspective, it can be seen in Figure 9 that students and teachers are a vital part of the three perspectives (social, organizational, and technological).

From the social perspective, most of the related actors are students, teachers, industry, organic units, digital platforms, government, teacher training units, information systems, or community.

From the organizational perspective, most of the related actors are students, teachers, organic units, university managers, business leader, content providers, rectory, or schools.

From the technological perspective, most of the related actors are students, teachers, university managers, DT team, faculty, or researchers.

Goals: The goals that have guided the DT processes at HEIs vary depending on the perspective (social, organizational, and technological) that that have been addressed.

From the social perspective, the goals that stand out are mainly those which seek to positively impact society, develop job skills, contribute to the growth and wellbeing of actors, improve HEIs credibility, afford the digital transformation of government, remove time and space barriers, and promote access to education, which positions the HEI human resource as a vital element to achieve DT, and to adapt and make curricula more flexible.

From the organizational perspective, the central goals of DT in HEIs that emerge are related to improving infrastructure, business process, administration, teaching, curricula, job, access, market openness, research, and digital marketing, as being novel aspects to consider.

From the technological perspective the major goals of the DT in HEIs that come out are related to provide technology to support human resource, teaching, innovation, administration, access, market openness, building process, society, and research.

Technology: The technology that has supported the DT at HEIs varies depending on the perspective (social, organizational, and technological) that have been addressed.

From the social perspective the main technologies that are taken into account in digital transformation processes are digital technology, social networks, learning management systems, big data, digital education tech, software, machine learning, computers, and RFID systems.

From the organizational perspective the technology that stand out are work management systems, business frameworks, digital technology, computers, and software.

From the technological perspective the most prominent technologies are digital educational tech, internet of things, data architecture, cloud computing, blockchain, mobile services, ecosystem of DT, virtual reality, business framework, and work management systems.

#### 4.2.3. Addressing of the DT in HEIs by the Different Actors

It is reasonable to understand that since the mission of the HEI is to educate, the main actors involved in the DT processes in the HEI are students and teachers. However, their needs and priorities vary depending on the perspective that has been addressed. The consolidation of every actor’s priorities varies and is detailed in the following paragraphs:

Student: The actor that has most influenced or forced HEIs to consider their own transition to a DT are their students. First, they expect to have the opportunity to study, without the barriers of time and space [3,20,25,29,30,33]. In consequence, HEIs should provide flexible curricula, digital learning, digital educational content, innovate teaching and research, personalized courses and experiences, and re-structure working processes [10,17,18,19,20,23,25,26,27,28,29,31,32,33]. Second, students expect to develop their capabilities and practical skills required in a digital world [20,23,26,27,29,31,32]. Similarly, students demand shorter individual certification programs where they can experience the potential of Industry 4.0 [14,17,23,31]. Third, students presuppose widen digital services offering at HEI level, facilitating the communication, collaboration, and co-creation of value in all the stakeholders [3,17,29]. This means the student expect the HEI to consider their experience as a student important [10,16,18,23,25] Finally, cost reduction will benefit the students’ economy [18,19,20,30,33].

Teacher: First, teachers should innovate their teaching, research, working process, and management experiences [10,14,19,23,27,28,29,31,33]. Second, teachers should impart digital services offered at a university level, to improve their productivity in teaching, plus facilitating the communication, collaboration, and co-creation of value of all the stakeholders [3,18,19,23,25,29,32].

Industry: First, HEIs should impart knowledge, competencies, and forecast which are needed for the industrial and economic complex of territories in the context of digitalization of the economy [23,31,32,33]. As a consequence, HEIs would provide new and innovative digital experiences, facilitating communication, collaboration, and co-creation of value in all the stakeholders [3,10,18,23,31,33]. Second, digital partnership is also a key item in their channels. Combining the data collected from digital partners and customer relationship can enable the institutions to create a predictive model with the help of technology [30]. Third, HEIs should provide shorter individual certification programs [17].

University managers: University managers realize in DT an opportunity to optimize process management within organic units and throughout all the University [3]. Similarly, they ensure the effective management of data transformation and digital adoption in business [14,32]. Thus, they improve data and information usage in all the decision support processes, either at an operational or strategic level, allowing decisions to be taken based in data and real data [3]. A challenge faced by university managers is related to the financial, technological constraints, and infrastructure capability to accommodate this DT of HEIs [10,34].

DT Team: It is important a strong leadership and a specialized team that can confidently explain and implement the DT in the HEI plan, understanding the magnitude of implementing digital work manager framework, an agile platform that could handle current, adaptive, and emergent processes. Moreover, DT Team should manage diverse cultural, behavioral, and operational forms of digital disruption [3,12,14,32].

Government: Public politics influence for modernization and streamlining of administrative processes and digital initiatives [3,18,27,30,32,33]. Forcing HEIs to speed up the development and adaptation of processes and services according to new societal requirements and legislative and regulatory changes, as well as organizational alterations [3,19,23].

Organic units: Organic units are directly benefited by DT in HEIs, since this transformation enables separate management and process execution from the physical place where the process is carried out [3]. Digital era requires self-managed teams in the working environment [22], becoming one of the challenges of organic units. 

Graduates: Graduates of higher education expect the HEIs to carry out activities related to the formation of competencies demanded by digital economy by providing skills in the field of IT at the world level and also new and innovative digital experiences [10,18,31,33].

Researcher: Researchers, authors and decision-makers have shown an increased interest in the causes and consequences of digitization for economies, states, and societies [23].

Community: New social requirements, legislative and regulatory changes, and organizational changes arise, leading to DT in HEIs accelerating its development and adaptation of processes and services [3,23,28].

Faculty: Faculty trust DT in HEIs permit the improvement of their productivity in teaching [19] and provide new and innovative digital experiences [10].

Digital platform: Digital platforms intervene as actors in the DT in HEI projects, as an enabler and support of this process. Digital workflow platform—enterprise architecture [3,12,18,19]; institutional framework to implement technology into teaching [18,23,27]; digital educational content-Repositories [17,20,23,29,32]; and e-learning systems [19,25,29].

IT business leader: IT Business leader actors lead re-engineering business processes, re-skilling people and transforming services by implementing digital initiatives in an integrated way with a framework to manage it-IT architecture management [3,10,17,20,23,30]. 

Teacher training unit: Teacher training unit becomes an important actor in the DT in HEI process due to the important challenge of digital literacy among all stakeholders [10,32] that require them to update their skills in the field at the global level, the principles of education in the areas of personalization, flexible design, and integration of various educational and labor trajectories [18,25,27,31,33].

Content providers: Content providers, as a key partner, can boost the quality and accessibility of contents provided by a university education [29,30].

Information system: Information system is key actor inside DT in HEIs improve data and information usage in all the decision support processes, either at an operational or strategic level, allowing decisions to be taken based in data and real data [3]. 

Academic department: Academic department leads to curriculum modernization, and administration processes [18,19,29,32].

Rectory: Rectory actor must be aware that DT is driven by business decisions and by business strategies [20] and, recognize heterogeneity of processes and practices to improve the overall efficiency of the University as a whole [3]. 

#### 4.2.4. Route Established by HEIs to Carry Out Their DT

The DT of a HEI is an organizational transformation that must be comprehensive and holistic, therefore the consolidation of the routes that have been established by higher education institutions to carry out their digital transformation and that are described in the papers is presented in the following paragraphs.

Guidelines DT in HEIs: From a macro-organizational vision, DT requires well planned digital strategy including the DT framework in which all key players and stakeholders can play an active role in shaping the university to thrive in the digital age [20]. For a digital strategy to be successful, it is necessary to ensure that the HEI has the necessary resources for its implementation [10]. Two aspects address DT: the transformation of products and services offered by organizations (improvement, expansion, and redefining) and the transformation of business processes for the provision of these products (creation, leverage, and integration) [17]. In terms of university teachers’ perspectives, technical as well as pedagogical guidance, is recommended [26]. Researchers [31] mentions the Institutes of Innovation as a very important infrastructure resource where training laboratories based on case studies can operate. 

DT center: Authors [17] stress that part of the implementation of DT in HEIs is that the university business process system requires the creation of a directory of administrative services for the training process and internal research, personnel management, infrastructure management, and other support services. 

Re-engineering process: In order to be successful, DT in HEIs require a profound re-engineering of all the supporting processes. This was a task that had to be dealt with profound sensitivity and attention, in order to overcome the natural resistance to change of the different schools, and due to the size of the university and the hundreds of processes managed daily, re-engineering and certification of all the processes and the de-materialization of document management, keeping at the same time the agility of the technologic infrastructures, is a very exigent process, requiring an innovative approach to succeed [16]. 

Build and running system: The re-engineering process goes hand in hand with building a system that supports the HEI business processes. Authors [16] divide this process into two stages. First, the platform should allow to dematerialize the full range of business processes of the University (several hundreds) in a relatively short time. Second, the WMS was expected to promote the harmonization, consolidation, and optimization of the working procedures. Furthermore, the software application architecture requires the creation of an integrated student life cycle management system, the integration of administrative information systems with systems of planning, and management of curricula and modules, databases of scientific data, and library repositories [17]. 

Competences center: A competence center is a strategic resource supporting the development of the human resources of the HEI. Digital capabilities are the capabilities which fit someone for the living, learning, and working in a digital society [20] and are the key enablers of university DT through the competent digital workforce [20]. 

Enterprise architecture–IT architecture management–digitalization: EA and business architecture constitute a conceptual tool that helps the organizations to understand their own structure and the approach by which they work [19]. Moreover, the implementation of the IT EA in the HEI settlements brought to light key benefits of integrating IT systems with educational processes in terms of increased agility of educational organizations, better decision making, and decreased IT related risks [35]. 

Change Management: To achieve the success of the execution of a project of the characteristics immersed in a DT in the HEI, it is vital [12] to minimize the potential negative effects of digital disruption, and the digital “syndromes” depends upon strong leadership and an understanding that disruption, transparency churn, and hypervigilance are to be expected during large-scale transformation.

### 4.3. Risks to Validity

First, the inclusion of all the relevant papers in the selection process is not guaranteed. This risk was mitigated by considering different scientific data bases in order to acquire the most relevant papers published by researchers. Second, once the eligibility phase was concluded, the number of papers was considerably reduced. The reason why this result was given, is that the search string through specialized databases provided papers that did not specifically guide their research to the DT in HEIs. In order to reduce this risk, authors manually changed the eligibility process, considering the researchers’ knowledge in the field. Third, the internal validity and the complexity of implementing the DT in HEIs, reduced the possibility of finding papers that holistically addressed the DT in HEIs and that answered all the research questions of this SLR. Consequently, to reduce this risk, authors analyzed each of the articles in detail, and applied the quality assurance checklist detailed in the quality assessment checklist in the inclusion section. Finally, the spreadsheet https://drive.google.com/file/d/17Ovoq4OibkzWFJMzxizonWGu2SmsTWcl/view?usp=sharing contains all the classification process that respond to the research questions of this SLR.

## 5. Conclusions

The DT in HEIs has been approached from the social, organizational, and technological aspects. 

The interest of HEIs to achieve their DT can be evidenced by the increase in articles that have been published in recent years (Figure 2). Furthermore, we have found that the tendency presents a defined importance from a social perspective, suggesting that researchers are aware of the importance of the human resource skills and capacities to successfully achieve the DT projects (Figure 3). 

The dimensions within a HEI that have been permeated by DT processes found in the literature are: teaching, infrastructure, curriculum, administration, research, business process, human resource, extension, digital transformation governance, information, and marketing (Figure 4). The foregoing measures the complexity that the DT process implies, and no article has included them all.

The authors have identified the actors who have been involved in the DT processes at HEIs, either because they have been leaders, or beneficiaries, and they are: students, teachers, industry, university managers, digital transformation team, government, organic units, alumni, researchers, community faculty, digital platforms, IT business leaders, teacher training units, parents, content providers, information systems, departments, schools, and rectory (Figure 9). The role these actors play depends on the dimension and perspective that has been addressed in the DT process (Figure 4, Figure 5, Figure 6, Figure 7 and Figure 8)

The multiplicity of ways in which DT has been addressed in HEIs have been reflected in each of the articles analyzed. This provides evidencing that DT in HEIs requires rethinking, restructuring, and reinventing, from its multi-purpose, multi-processes, multi-disciplinary, multi-state, and multi-actoral character. It is a collective effort that places the person in the center of the process of development, transformation, and its impact on society. That is, DT should be an integral and holistic transformation of the HEI.

This review suggests that the DT dimensions inside HEIs do not just imply technological progress, instead it is more transcendental, and generates changes of meaning, affecting the culture immersed in the university, the administrative, formative activities and their evaluations, the pedagogical approaches, the teaching, research, extension and administrative processes, as well as the people immersed in it. 

Most of the papers’ main efforts focused DT in HEIs and it has been approached in a segmented way, evidencing the absence of adoption methodologies of these kind of initiatives at the holistic HEI level that align with the business model, operational processes, and user experience, taking into account internal digital capabilities and its own prospective view.

Research on conceptualization and methodologies to adopt DT in HEIs should be deepened.

## Figures and Tables

**Figure 1 sensors-20-03291-f001:**
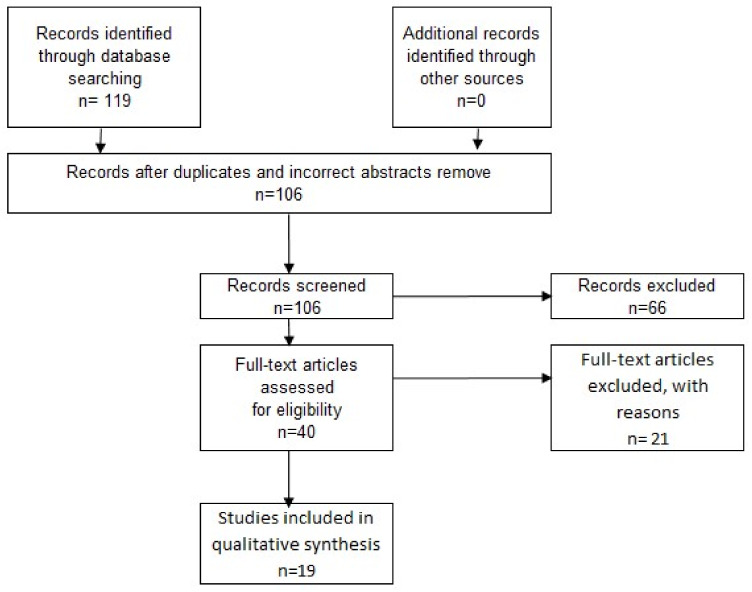
Summary review protocol.

**Figure 2 sensors-20-03291-f002:**
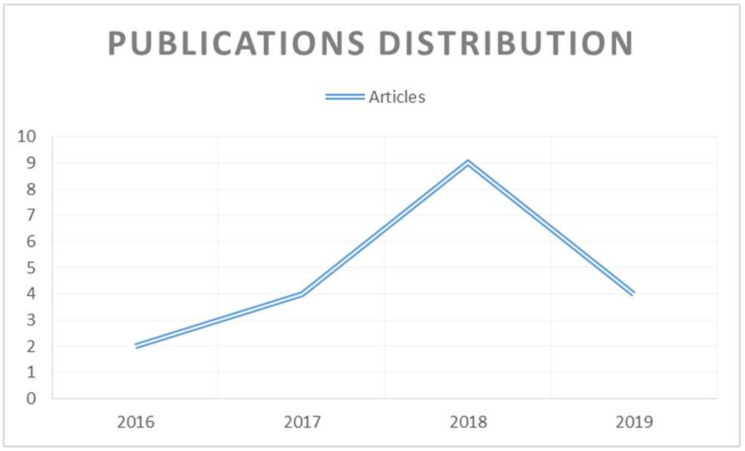
Publications distribution.

**Figure 3 sensors-20-03291-f003:**
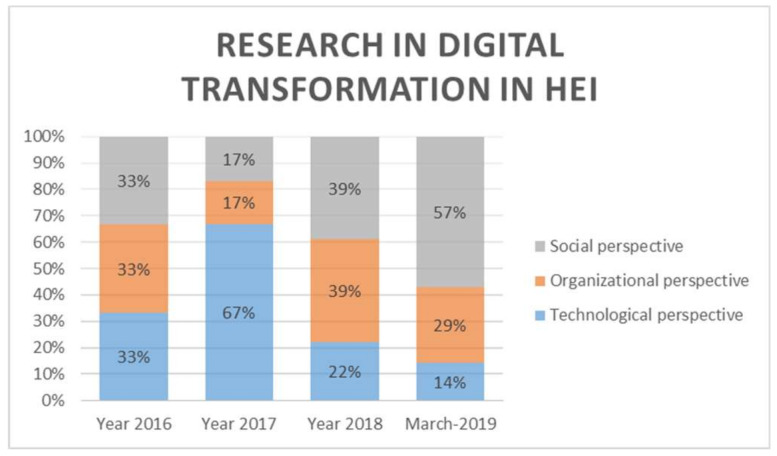
Research in digital transformation in HEIs.

**Figure 4 sensors-20-03291-f004:**
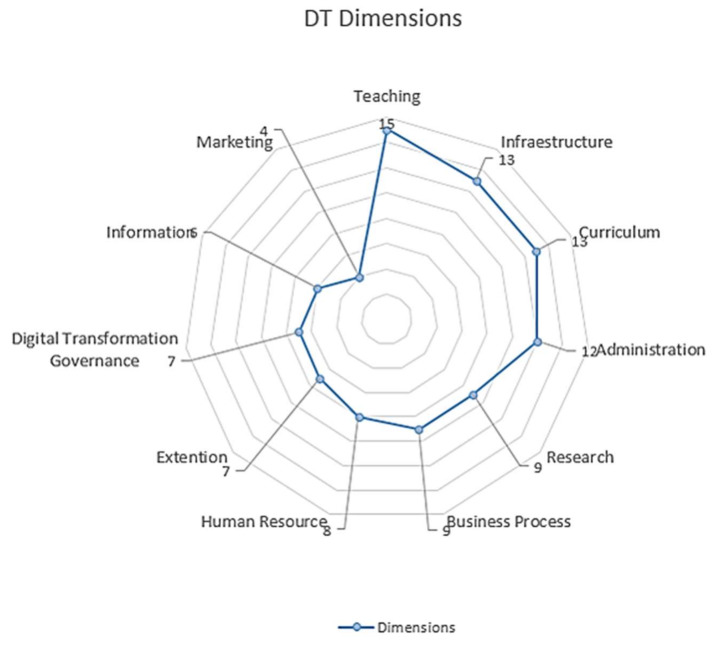
Radar of the dimensions of the DT in HEIs.

**Figure 5 sensors-20-03291-f005:**
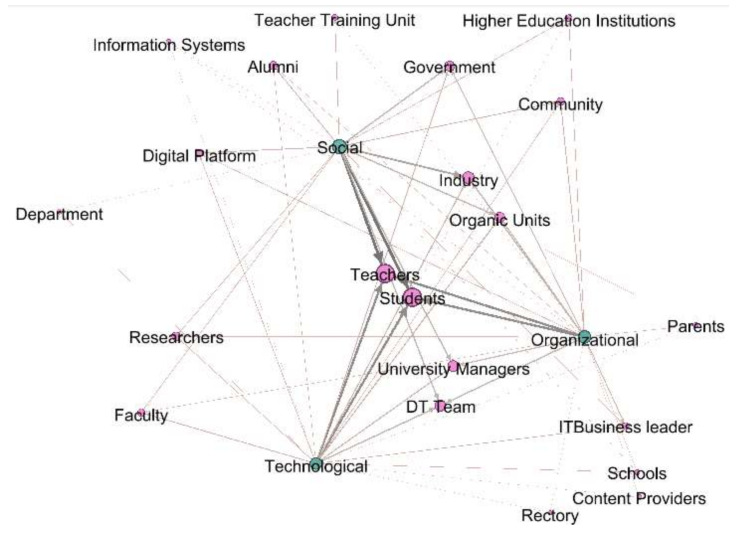
Actors of DT in HEIs.

**Figure 6 sensors-20-03291-f006:**
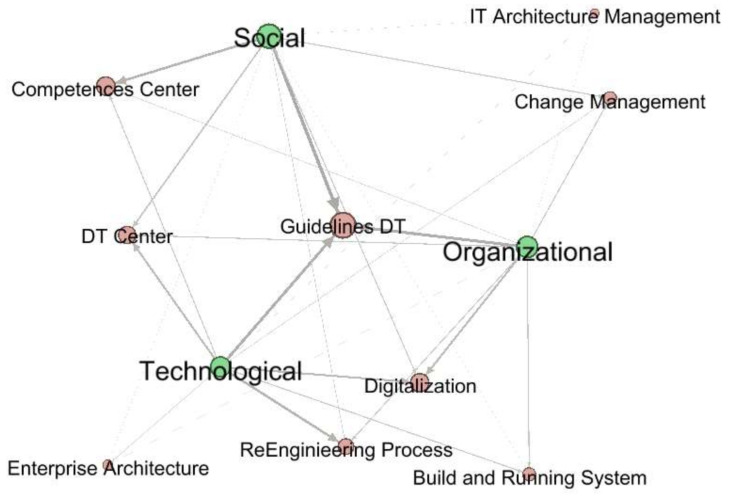
Methods applied in DT in HEIs.

**Figure 7 sensors-20-03291-f007:**
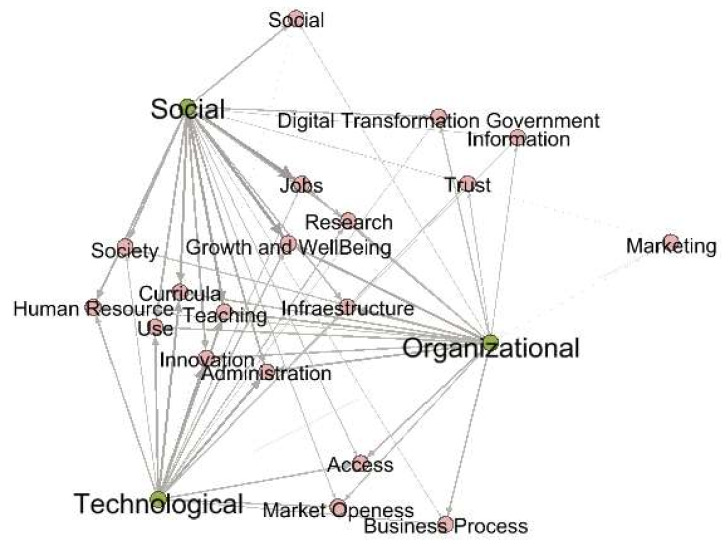
Goals of DT in HEIs.

**Figure 8 sensors-20-03291-f008:**
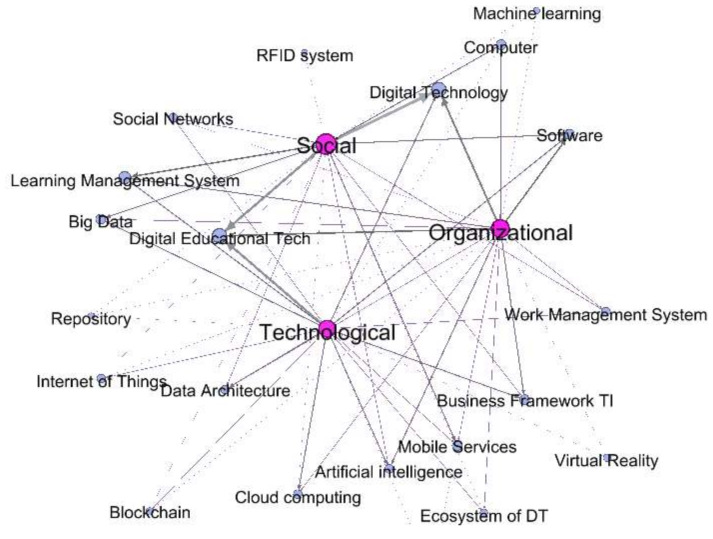
Technologies used in DT in HEIs.

**Figure 9 sensors-20-03291-f009:**
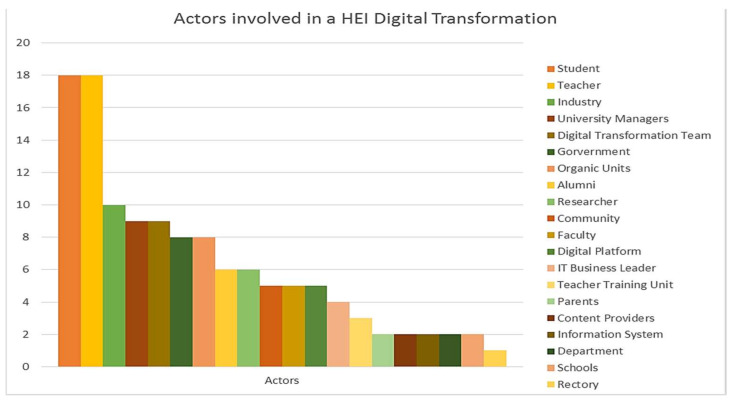
Actors involved in a DT at a HEI.

**Figure 10 sensors-20-03291-f010:**
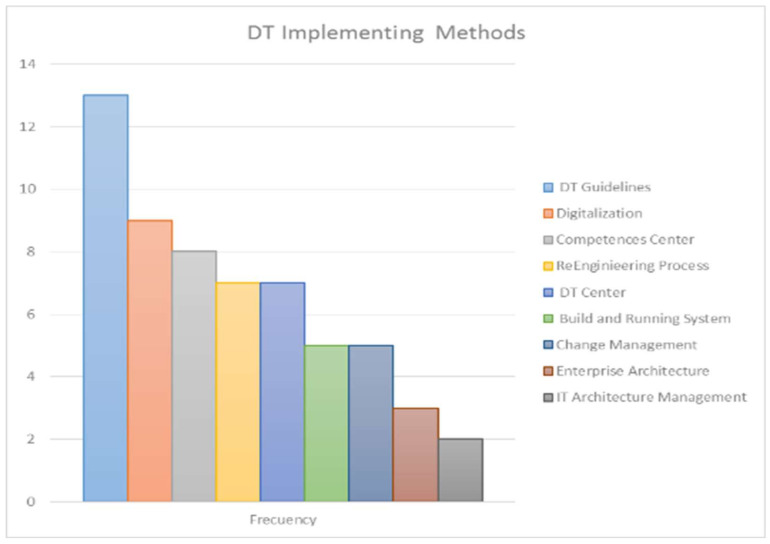
Implementing methods of DT at a HEI.

**Table 1 sensors-20-03291-t001:** Quality assessment checklist.

Level	Description	Score
Yes	Information is explicitly defined/evaluated	1
Partially	information is implicit/stated	0.5
No	information is not inferable	0

**Table 2 sensors-20-03291-t002:** Acronyms to classify information.

Source	Acronym
DT description	Technological (TC) Organizational (OR) Social (S). We use the classification proposed by [9]
DT Goals and services	Use (US) Access (AC) Innovation (IN) Jobs (JO) Society (SO) Trust (TR) Market Openness (MO) Growth and Wellbeing (GW)
DT dimensions and characteristics	Research (RE) Teaching (TE) Social (SO) Business Process (BP) Human Resource (HC) Curricula (CU) Infrastructure (IN) DT Government (DG) Administration (AD) Marketing (MK) Information (INF)
Actors or Stakeholders involved in DT	Students (S) Alumni (A)- Teachers (T) -Researchers (R) University Managers (M) Community (C) Faculty (F) Department (D) Government (G) IT Business leader (ITB) Rectory (Ry) Organic Units (OU) Schools (Sc) DT Team (Te) Teacher Training Unit (TT) Industry (I) Parents (P) Content Providers (CP) HEIs (HEI) Digital Platform (DP) Information Systems (IS) Library (L)
DT implementing methods	Guidelines DT (G) DT Center (DC) Reengineering Process (RE) Build and Running System (BS) IT Architecture Management (ITAM) Competences Center (CC) Digitalization (DI) Change Management (CM) Enterprise Architecture (EA)
Technologies used	Work Management System (WMS) Enterprise Resource Planning (ERP) Business Framework TI (BF) Information Communication Technology (ICT) Software (SW) Learning Management System (LMS) Digital Educational Tech (DE) Computer (PC) Cloud computing (CL) Blockchain (B) Internet of Things (IoT) Mobile Services (MS) Big Data (BD) Social Networks (SN) Data Architecture (DA) Digital Technology (DT) Ecosystem of DT (ECO) Computer Power 5G Networks, Artificial intelligence (AI) Virtual Reality (VR) Augmented Reality (AR) RFID system Machine learning (ML) Repository (Re)
Governance	Public Politics (PP) Governability for DT (GDT)

**Table 3 sensors-20-03291-t003:** Records obtained.

Criteria	Filters	Scopus	Web of Science (WoS)
Restriction	Topic (title, abstract, author keywords)	129	31
Period	2001–2019. The first article published in WoS was in 2001 1980–2019 Scopus	128	30
Document type	Articles and conference proceedings	107	30
Language	English	100	19
Total		119	

**Table 4 sensors-20-03291-t004:** Number of elected papers.

Criteria	Papers
Articles Elected	40
Excluded articles	21

**Table 5 sensors-20-03291-t005:** Full reading papers included.

Criteria	Papers
Full reading papers	19
Excluded articles	21

**Table 6 sensors-20-03291-t006:** Definitions related to DT in HEIs.

Paper	Digital Transformation
[10]	DT is fundamentally about change and it involves people, processes, strategies, structures, and competitive dynamics [11].
[12]	Digital disruption is defined as the changes facilitated by digital technologies that occur at a pace and magnitude that disrupt established ways of value creation, social interactions, doing business and more generally our thinking [13].
[14]	The realignment of, or new investment in, technology and business models to more effectively engage digital customers at every touch point in the customer experience lifecycle. Companies needed to think of DT as a “formal effort to renovate business vision, models, and investments for a new digital economy [15].”
[16]	DT goes well beyond de-materialization of processes, encompassing an innovative use of new technologies (cloud, social, mobile, and analytics) to promote new services, re-define business models, and innovative interactions with its users.
[17]	DT of the university education system should have a broader focus and must include the modernization of corporate IT architecture management, which could provide an important contribution to structuring the efforts of innovation in education.
[18]	The modern developments in the area of modernizing educational system with the aid of ITC technology and applied process thinking principles in the attempt to capture and model interrelated activities required to integrate digital technologies in teaching, learning, and organizational practices.
[19]	DT is an accelerated evolution. It is also revolution because of its radical and structural implications for people as for infrastructure that also requires new educational and business models.
[20]	Digital business transformation can be defined as the modification of business processes, procedures, capabilities and policies to take advantage of the changes and opportunities presented by new digital technologies, as well as the impact they have on society, while always thinking about current and future trends.

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
