# Peer review of "Digital Transformation in Higher Education Institutions: A Systematic Literature Review"

_sensors, 2020, doi:10.3390/s20113291_

Round 1

Reviewer 1 Report

The authors follow all reviewers comments.

Reviewer 2 Report

The abstract narrates the summarized beforehand knowledge of the article in a good manner.
The introduction part often is crucial in gaining the attention of the reviewers and readers in the topic under discussion.
Author has managed to put his efforts in gaining that interest. Up to the mark in terms of having a well discussed
conclusion.
It is my fair judgment that this article is fit to be published without any need of change in it.

This manuscript is a resubmission of an earlier submission. The following is a list of the peer review reports and author responses from that submission.

Round 1

Reviewer 1 Report

This paper presents a systematic review of digital transformation in higher education institutions which is not in the frame of Sensors (as a discipline) nor in the field if sensors research. The authors have to check the scope of Sensors journal. This paper might be published in a paper of educational, managerial or an organizational journal. In addition, please check the whole manuscript with native English.

Author Response

The authors appreciate your comments.    

Reviewer 2 Report

The paper "Digital Transformation in Higher Education 2 Institutions: A Systematic Literature Review" is well written and structured. The main goal is to determine the dimensions, actors, implementations that have taken place in the process of Digital Transformation within Higher Education Institution through a systematic literature review.

The authors identify the problem accurately and have chosen the correct investigation methodology. The results obtained are interesting, but they need further discussion.

It is necessary to explain the process used to obtain the graphs in figures 2, 3, 4 and 5.

The conclusions must be more explicit and objective.

Author Response

Authors have been major revisions to the paper “Digital Transformation in Higher Education Institutions: Systematic Literature Review” according to the requests that reviewers have made in the evaluation process.

The detail of the revisions that have been made are described in the following tables

Reviewer 2

Suggestion

Adjustments

pages

“methodology. The results obtained are interesting, but they need further discussion”

Section 4 Discussion has been extended according with the results.

14-20

It is necessary to explain the process used to obtain the graphs in figures 2, 3, 4 and 5.

The description of the creation of the graphs has been expanded. The Spreadsheet link that contains the standardized information executed by Giphi Software was include. Section 3.4. Interrelationships inside DT of HEIs-MQ2.1.

10

The conclusions must be more explicit and objective

Conclusions have been adjusted to be explicit and objective. Section 5. Conclusions

20

To view the document without revisions, in Word the option Review / Change Control / "No revisions" must be selected.

Sincerely,

Lina María Castro Benavides

National University of Colombia

Sede Manizales

Campus La Nubia

e-mail: licastrob@unal.edu.co

Reviewer 3 Report

The topic is interesting but I found several weaknesses. Please see below:

  • Section 1.1 has two references and therefore is not well supported. It is not clear to me the problem you are trying to solve/help or the motivation
  • Section 1.2 has too much RQ/MQ for an SLR in my honest opinion. This may be a result of a weak introduction.
  • Section 1.3 do not support the motivation. The absence of a study do not necessarily means that there is a gap to fulfil. You need to justify why such gap should be fulfilled. And that is not present.
  • Section 2 could be improved with the adoption of a strongest SLR methodology. You use PRISMA but is not enough in my opinion. Please see for example the work of Barbara Kitchenham or other similar. I do not know which data bases you have used or which is the objective of the SLR.
  • I do not understand the refered concepts in Table 5. You argue that such concepts are aligned with the RQ/MQ but the serach strig do not refer such concepts at all. This reinforced my opinion that you have too much RQ/MQ and also indicates that the methodology is not well applied.

Therefore, With a fair methodology adoption I do not trust in the results and therefore I'm sorry but my decision is against the publication of this article.

Author Response

Authors have been major revisions to the paper “Digital Transformation in Higher Education Institutions: Systematic Literature Review” according to the requests that reviewers have made in the evaluation process.

The detail of the revisions that have been made are described in the following table.

Reviewer 3

Suggestion

Adjustments

pages

·         Section 1.1 has two references and therefore is not well supported. It is not clear to me the problem you are trying to solve/help or the motivation

Problem and motivation have been supported in the Introduction Section with additional literature.

Section 1.1. Rational

1,2

Section 1.2 has too much RQ/MQ for an SLR in my honest opinion. This may be a result of a weak introduction

In accordance with the adjustments made to the Introduction Section, the RQ / MQ taken into account in the investigation have been supported. 1.2. Review Questions.

2

Section 1.3 do not support the motivation. The absence of a study do not necessarily means that there is a gap to fulfil. You need to justify why such gap should be fulfilled. And that is not present

In section 1.3 the motivation for conducting this research has been expanded

3

Section 2 could be improved with the adoption of a strongest SLR methodology. You use PRISMA but is not enough in my opinion. Please see for example the work of Barbara Kitchenham or other similar. I do not know which data bases you have used or which is the objective of the SLR.

The entire document has been adjusted following the Barbara Kitchenham Methodology.

Sections 2, 3, 4, 5

3-21

·         I do not understand the refered concepts in Table 5. You argue that such concepts are aligned with the RQ/MQ but the serach strig do not refer such concepts at all. This reinforced my opinion that you have too much RQ/MQ and also indicates that the methodology is not well applied.

2.5.2. Data extraction procedures Section was included to explain how these information classification codes were obtained.

5,6

To view the document without revisions, in Word the option Review / Change Control / "No revisions" must be selected.

Sincerely,

Lina María Castro Benavides

National University of Colombia

Sede Manizales

Campus La Nubia

e-mail: licastrob@unal.edu.co

Round 2

Reviewer 1 Report

This paper presents a systematic review of digital transformation in higher education institutions which is not in the frame of Sensors (as a discipline) nor in the field if sensors research. As I have mentioned, this paper is not included the scope of Sensors journal and might be submitted in educational, managerial, or organizational journals.

Reviewer 2 Report

The paper new version included all my recommendations.

Reviewer 3 Report

I recognize your effort to evolve the article.

Nevertheless, IMHO your SLR is not well done. Too much RQ, wich are not well supported and therefore your motivation is not clear. As a result, conclusions are not well aligned with  RQ as well.

Please, in the future, do not send a revised version with trackchanges.